# Automated PD-L1 Scoring Using Artificial Intelligence in Head and Neck Squamous Cell Carcinoma

**DOI:** 10.3390/cancers13174409

**Published:** 2021-08-31

**Authors:** Behrus Puladi, Mark Ooms, Svetlana Kintsler, Khosrow Siamak Houschyar, Florian Steib, Ali Modabber, Frank Hölzle, Ruth Knüchel-Clarke, Till Braunschweig

**Affiliations:** 1Department of Oral and Maxillofacial Surgery, University Hospital RWTH Aachen, 52074 Aachen, Germany; bpuladi@ukaachen.de (B.P.); mooms@ukaachen.de (M.O.); amodabber@ukaachen.de (A.M.); fhoelzle@ukaachen.de (F.H.); 2Institute of Pathology, University Hospital RWTH Aachen, 52074 Aachen, Germany; skintsler@ukaachen.de (S.K.); fsteib@ukaachen.de (F.S.); rknuechel-clarke@ukaachen.de (R.K.-C.); 3Institute of Medical Informatics, University Hospital RWTH Aachen, 52074 Aachen, Germany; 4Department of Dermatology and Allergology, University Hospital RWTH Aachen, 52074 Aachen, Germany; khosrow-houschyar@gmx.de

**Keywords:** PD-L1 scoring, head and neck squamous cell carcinoma, deep learning, tumor detection, medical image analysis, open-source

## Abstract

**Simple Summary:**

Immunotherapy forms an emerging and successful field in cancer therapy using checkpoint inhibitors (e.g., anti PD-L1, anti PD-1), preventing immune escape of the tumor. However, these drugs are often only effective in a subpopulation of patients. To identify such patients, various so-called PD-L1 scores based on PD-L1 expression by immunohistochemistry in tumor tissue had been established. However, these scores may vary between different human investigators, which could negatively influence treatment decisions. The aim of our work was to obtain reproducible and reliable PD-L1 scores using artificial intelligence. Our results show comparable performance between human-human and human-machine interactions and could provide a deeper insight into the function and limitations of automated scoring by artificial intelligence. This could serve as a basis to improve patient selection for checkpoint inhibitors in the future.

**Abstract:**

Immune checkpoint inhibitors (ICI) represent a new therapeutic approach in recurrent and metastatic head and neck squamous cell carcinoma (HNSCC). The patient selection for the PD-1/PD-L1 inhibitor therapy is based on the degree of PD-L1 expression in immunohistochemistry reflected by manually determined PD-L1 scores. However, manual scoring shows variability between different investigators and is influenced by cognitive and visual traps and could therefore negatively influence treatment decisions. Automated PD-L1 scoring could facilitate reliable and reproducible results. Our novel approach uses three neural networks sequentially applied for fully automated PD-L1 scoring of all three established PD-L1 scores: tumor proportion score (TPS), combined positive score (CPS) and tumor-infiltrating immune cell score (ICS). Our approach was validated using WSIs of HNSCC cases and compared with manual PD-L1 scoring by human investigators. The inter-rater correlation (ICC) between human and machine was very similar to the human-human correlation. The ICC was slightly higher between human-machine compared to human-human for the CPS and ICS, but a slightly lower for the TPS. Our study provides deeper insights into automated PD-L1 scoring by neural networks and its limitations. This may serve as a basis to improve ICI patient selection in the future.

## 1. Introduction

Tumors of the head and neck are responsible for about 4.6% of all cancer cases with an estimated incidence of 834,860 reported cases worldwide, and for 4.5% of all cancer deaths with about 431,131 reported deaths in 2018 [1]. Squamous cell carcinoma is the predominant histologic type in head and neck tumors, accounting for nearly 90% of cases [2]. The treatment of choice for HNSCC of the oral cavity and advanced stages of the larynx and hypopharynx is usually surgery. Depending on the anatomic location and stage, radio- and/or chemotherapy may also be required. In advanced stages or recurrent or metastatic cancer cases primary radio- and/or chemotherapy is applied [3]. Nearly 50% of the patients die from the disease depending on stage, anatomic location, and other factors and 50% of deaths occur in the first two years [4]. Recurrent and metastatic HNSCC (R/M HNSCC) has an even poorer prognosis with a median survival time between 6 and 12 months [5].

Immunotherapy with PD-L1 checkpoint inhibitors offers a novel and promising immune modulating treatment for unresectable R/M HNSCC stages [6]. Programmed death ligand 1 (PD-L1), on e.g., tumor cells, regulates the cytotoxic activity by binding to the inhibitory programmed death receptor 1 (PD-1) of cytotoxic T-cells, leading to deactivation of these T-cells and allows an immune escape of tumor cells [7]. By interrupting this mechanism, checkpoint inhibitors such as pembrolizumab (Keytruda^®^, MSD, Kenilworth, NJ, USA) or nivolumab (Opdivo^®^, BMS, New York, NY, USA) bind to PD-1 with the aim to prevent an immune escape of tumor cells leading to a prolonged progression-free status and overall survival [6]. However, this therapeutic approach appears to be most effective in a subpopulation of patients, depending on the extent of PD-L1 expression in tumor cells and tumor-infiltrating immune cells [8,9].

Generally, for patient selection for (often first-line) anti-PD-1/anti-PD-L1 checkpoint-inhibitor therapy, so-called PD-L1 scores were developed. Over the last four years, three PD-L1 scores were established, which have to be determined manually by pathologists based on the immunohistological expression of PD-L1 in tumor cells and/or infiltrating immune cells in tumor main specimens or biopsies [6,10]. The tumor proportion score (TPS) is defined by the percentage of all tumor cells presenting membranous PD-L1 expression in immunohistochemistry. In contrast, immune cell score (ICS) describes the percentage of the area of immunohistochemically PD-L1-positive infiltrating immune cells in respect to the total area of the tumor tissue. The combined positive score (CPS) reflects the proportion of the number of positively stained tumor cells and infiltrating immune cells in relation to the number of tumor cells and is defined by a maximum value of 100 [10,11].

Of currently approved drugs, pembrolizumab is as a single-agent approved by the FDA and the EMA for first-line therapy of unresectable R/M HNSCC showing a CPS ≥ 1 [12,13]. Furthermore, it is approved by the EMA in combination with platinum and fluorouracil chemotherapy for unresectable R/M HNSCC with a CPS of ≥ 1 and as a monotherapy for R/M HNSCC with a TPS of ≥50% under or after platinum-containing chemotherapy [13]. In addition, pembrolizumab showed additional benefit in studies with PD-L1 high-expressors with a CPS ≥ 20 in R/M HNSCC [6]. In contrast, ICS is not currently used in treatment decisions for pembrolizumab and R/M HNSCC, although it is used for atezolizumab (Tecentriq®, Roche, Basel, Switzerland) in other tumor entities such as triple-negative breast cancer (ICS ≥ 1%) and urothelial carcinoma (ICS ≥ 5%) [14,15].

PD-L1 scoring was complicated in the various studies performed by the presence of different testing systems based on individual PD-L1 antibody clones (e.g., 22C3, SP142, 28-8, SP263, etc.), testing platforms (e.g., Ventana/Roche, Roche, Basel, Switzerland), DAKO/Agilent, Agilent, Santa Clara, CA, USA), immunotherapeutic agent chosen (pembrolizumab, nivolumab, durvalumab, atezolizumab, etc.), and validity for specific tumor entities, finally leading to different cutoff values to conduct therapy decisions [16,17].

As a result, reproducibility of PD-L1 scores based on different PD-L1 antibody clones and investigators has been the subject of various studies, showing varying degrees of comparability, interchangeability, and inter-observer reliability [18,19,20,21]. Additionally, harmonization studies were conducted concerning different PD-L1 antibody clones and different observers and thus make PD-L1 scoring more comprehensible and reliable [22,23,24].

The effectiveness of artificial intelligence (AI) in medical image analysis (IA) of histopathological images has been widely demonstrated [25], suggesting that automated determination bases on AI/IA could enable reliable and reproducible PD-L1 scoring [14]. Attempts have already been made to determine PD-L1 scores using AI/IA [16,26,27,28,29,30,31,32,33,34,35,36]. However, an open-source and automated determination of all three PD-L1 scores simultaneously in whole slides images (WSI) has not yet been described in the available literature.

In the presented study we propose a novel approach using three consecutive distinct neural networks and methods of image analysis to perform the fully automated determination of the three PD-L1 scores (TPS, CPS, and ICS) in WSIs. Furthermore, the performance of our approach was evaluated by comparing machine-based PD-L1 scores with manually determined PD-L1 scores.

## 2. Materials and Methods

### 2.1. Tissue Training Data, Sampling and Immunohistochemistry

Training data for the first neural network for automated tumor detection of HNSCC and the third neural network for classifying cells into tumor, immune, and stromal cells were obtained from WSIs of the public tumor database CPTAC (https://wiki.cancerimagingarchive.net/display/Public/CPTAC-HNSCC, accessed on 24 August 2021). For this purpose, different tissue classes in CPTAC WSIs were annotated using QuPath and exported as image tiles, each with an edge length of 256 µm, by a script. The training data were independently controlled by two authors (B.P. and T.B.). Instead for the second neural network for nucleus recognition, an already pre-trained and public available network was used (https://github.com/stardist/stardist-imagej/tree/master/src/main/resources/models/2D/he_heavy_augment.zip, accessed on 24 August 2021). The use of public available WSIs from CPTAC was approved by the ethic-commission of the University Hospital RWTH Aachen (approval number 20-567).

To evaluate the presented approach of automated PD-L1 scoring, 54 formalin-fixed, paraffin-embedded, anonymized samples of solid HNSCC from the Institute of Pathology of the University Hospital RWTH Aachen were randomly and retrospectively selected, approved by the ethic-commission of the University Hospital RWTH Aachen (approval number 20-200). Sections were immunohistochemically stained in an autostainer (Link48, Agilent) by anti-PD-L1 antibody (clone 22C3, Agilent, 1:50) and Flex detection kit (DAKO, Agilent, Santa Clara, CA, United States) according to manufacturer recommendations and counterstained with hematoxylin. After dehydration and coverslipping, sections were scanned at 40× (equiv. 400× in light microscopy) magnification (NanoZoomer 2.0-HT, Hamamatsu Photonics, Hamamatsu, Japan).

### 2.2. Automated PD-L1 Scoring

#### 2.2.1. Tumor Detection

The first neural network for automated tumor detection in WSIs of tumor sections was realized using a convolutional neural network (CNN). For this purpose, the CNN “shufflenet” was chosen and trained [37]. Eight tissue classes were selected (adipose tissue, cartilage, squamous epithelium, glandular tissue, immune cells, muscle tissue, tumor tissue, stromal tissue) and served as training data with 40,000 tile images (5000 tile images per tissue class, size 256 µm × 256 µm) previously extracted from hematoxylin-eosin-stained WSIs of the CPTAC tumor database. After dividing color channels, only the hematoxylin channel was used for both training and detection as our test tissue sections were stained with DAB (PD-L1 22C3 clone) and counterstained with hematoxylin. The trained CNN was finally capable of detecting tumor tissue of HNSCC and seven different non-tumorous tissue types.

Our test WSIs were then automatically annotated for tumor tissue by the CNN (Figure 1A). For this purpose, the entire WSIs were traversed in a sliding window manner and divided into individual image tiles, which were classified and exported as a binary labelmap. To avoid defragmented annotations, very small binary labelmap areas and gaps in the binary labelmap were removed. Finally, the binary labelmap was imported into QuPath [38] as an annotation [39] and the color background was normalized (see “Estimate_background_values.groovy” in Reference [38]).

To visualize the detectable features of our trained CNN the “DeepDream” method was used (Figure 2A) and implemented using the Deep Learning Toolbox in MATLAB [40]. The “Deep Dream” method can be used to visualize abstractions of the recognizable image classes of a CNN [41], such as the above-mentioned histological tissue classes.

#### 2.2.2. Cell Detection and Classification

To facilitate cell detection in previously annotated tumor regions, a second neural network, an already trained model of the CNN “StarDist” was used [42] (Figure 1B).

The resulting cell-specific raw measurements (such as cell size, nucleus size, optical density of hematoxylin staining of the nucleus, etc.) were additionally supplemented by smoothed features by calculating the weighted mean of the corresponding measurement of neighboring cells. This helps to improve the subsequent cell classification [43].

Following this, a third neural network classified the recognized cells into three classes important for the scoring of immunohistochemistry staining (tumor, immune, and stromal cells) (Figure 1C). Stromal cells were selected as the third class since they frequently surround tumor tissue and should be excluded for scoring. For this purpose, a multilayer perceptron (MLP) was used [44], which was previously trained on detected cells in montaged images (12 × 12 tiles) and tested on equivalent montaged test images from the CPTAC training data set (Figure 2C). To reveal the deciding morphological and color characteristics in cell classification, the importance variables for the MLP were determined [45].

#### 2.2.3. Detecting PD-L1 Positive Cells and Calculating PD-L1 Scores

Positive tumor and immune cells were detected with an optical density (OD) threshold in color channel selected DAB staining previously defined in literature [26,28,38]. This threshold was used for tumor cells for membranous staining only and for immune cells for total cell staining (Figure 1D).

Finally, PD-L1 scores (TPS, CPS, ICS) were calculated according to the following formulas:(1)$$\mathrm{TPS}\;=\;\frac{\mathrm{No}.\;\mathrm{PD}\text{-}L1\text{-}\mathrm{stained}\;\mathrm{tumor}\;\mathrm{cells}}{\mathrm{No}.\;\mathrm{of}\;\mathrm{viable}\;\mathrm{tumor}\;\mathrm{cells}}\times \;100$$
(2)$$\mathrm{CPS}\;=\frac{\mathrm{No}.\;\mathrm{PD}\text{-}L1\text{-}\mathrm{stained}\;\mathrm{tumor}\;\mathrm{cells}+\mathrm{No}.\;\mathrm{PD}\text{-}L1\text{-}\mathrm{stained}\;\mathrm{immune}\;\mathrm{cells}}{\mathrm{No}.\;\mathrm{of}\;\mathrm{viable}\;\mathrm{tumor}\;\mathrm{cells}}\times 100$$
(3)$$\mathrm{ICS}\;=\frac{\mathrm{Area}\;\mathrm{of}\;\mathrm{PD}\text{-}L1\text{-}\mathrm{stained}\;\mathrm{immune}\;\mathrm{cells}}{\mathrm{Area}\;\mathrm{of}\;\mathrm{the}\;\mathrm{tumor}}$$

#### 2.2.4. Used Soft- and Hardware

Tumor detection was implemented using the MATLAB programming language and the Deep Learning Toolbox [40]. Cell detection with CNN “StarDist” and classification with the MLP and subsequent calculation of scores was performed using the groovy programming language in the open-source software QuPath [38]. Automated PD-L1 scoring was performed on the same workstation (CPU: Ryzen 5950X, Memory: 128 GB, GPU: RTX 3090). The source code used is available in the Appendix A.

### 2.3. Manual PD-L1 Scoring

For all 54 HNSCC slides the PD-L1 scores TPS, CPS and ICS were determined manually and blinded by four human investigators (three trained pathologists: T.B., F.S., S.K.; and after instruction B.P.) using a standard brightfield microscope according to official PD-L1 evaluation manual for sections stained with the antibody IHC 22C3 [11]. The PD-L1 scores were calculated according to the above-mentioned Equations (1)–(3). Based on results of the four investigators, average PD-L1 scores were calculated for each case.

### 2.4. Statistical Analysis

R programming language was used for statistics. A *p*-value < 0.05 was considered statistically significant. To evaluate the inter-observer reliability of PD-L1 scoring, intraclass correlation coefficient (ICC) was calculated using the R package “irr” [46] and determined for the PD-L1 scores between the four human investigators and between the mean of the four human investigators and the automated PD-L1 scores. An ICC of <0.5 means poor, of 0.5–0.75 moderate, of 0.75–0.9 good, and of >0.9 excellent reliability [47]. Normal distribution was tested using the Shapiro–Wilk test and, thus, a systemic difference was tested using the Wilcoxon test between human and machine. Concordance between machine and human mean was calculated using a TPS cutoff of ≥1% and ≥50%, a CPS cutoff of ≥1 and ≥20, and an ICS cutoff of ≥1% and ≥5%, as described in the literature [6,14]. Concordance for cutoffs (ordinal data) were calculated using weighted kappa (κ) with equal weights. For more than two raters, the arithmetic mean of the pairwise estimates of all pairs was used [48]. κ < 0.00 was rated as poor, 0.00 ≤ κ ≤ 0.20 as mild, 0.21 ≤ κ ≤ 0.40 as moderate, 0.41 ≤ κ ≤ 0.6 as moderate, 0.61 ≤ κ ≤ 0.8 as substantial, and κ ≥ 0.80 as near perfect [49]. The importance variables were determined using the R package “NeuralNetTools” [50]. 

## 3. Results

### 3.1. Evaluation of Tumor Detection and Cell Classification by Neural Networks

The first neural network (CNN “shufflenet”) for annotation of tumor tissue in WSIs was able to identify eight different tissue classes. The “DeepDream” method was able to generate eight of recognizable tissue classes. This enabled humans to verify that the understandable morphological decision features were the basis for classification. For example, in the Deep Dream image “EPI” (Figure 2A), a basal lamina and stratified cells can be easily recognized, as would be expected in epithelium. To verify the performance of our first network, a CPTAC test set of 6000 tile images (750 tile images per tissue class) was used to determine the prediction performance. For all tissue classes, the correct class could be assigned in 96.8% of the tiles and for tumor tissue in 94.6% of the cases (Figure 2B).

The second neural network (CNN “StarDist”) was not re-evaluated as part of this study as performance using the 2018 Data Science Bowl dataset has already been described in the literature and would be outside the scope of this study [42].

The performance of the third neural network (MLP) was validated using 30,000 cells (normalized to 10,000 cells per class) in CPTAC tile images of tumor, immune, and stromal tissue. 88.0% of tumor cells, 99.9% of immune cells and 87.1% of stromal cells could be correctly classified by their morphological characteristics. The overall accuracy of the cell classification was 97.2% (Figure 2C). In MLP decision making circularity and minimum diameter of cells and nuclei (µm), area of cells (µm^2^), solidity of nuclei and nucleus/cell area ratios were the most important morphological variables. For the OD of hematoxylin stain, dispersion (OD) of cells and nuclei, median (OD) of membrane and cell, max (OD) of cell, cytoplasm and nucleus, and min (OD) of membrane and nucleus were most important color variables. Both raw and smoothed features were important for classification into different cell types (Figure 2D,E). In this context, “smoothed” implies a weighted mean of the corresponding measurements of neighboring cells for an improved cell classification.

The three classified cell classes (tumor, immune, and stromal cells) differed significantly in their morphological characteristics (cell/nuclear shape and size, hematoxylin staining (OD)) and were well reflected by the third network in our PD-L1 HNSCC test dataset (Appendix A).

### 3.2. Comparison of Automated and Manual PD-L1 Scores

To test our automated PD-L1 scoring, three different PD-L1 scores were determined for 54 cases of HNSCC based on PD-L1 assessment manually by the above mentioned four human investigators (T.B., S.K., F.S., B.P.) and automated by machine (Figure 3A–C, in pseudo-log scale for better visualization of low scores see Appendix A).

The inter-rater reliability of PD-L1 scores between the four investigators was excellent for the TPS (ICC = 0.91; 95% CI 0.86–0.94, *p* < 0.001), moderate for the CPS (ICC = 0.71; 95% CI 0.57–0.81, *p* < 0.001), and poor for the ICS (ICC = 0.26; 95% CI 0.12–0.41, *p* < 0.001) (Figure 4A). Concordantly, inter-rater reliability between the average PD-L1 scores of the four investigators and the machine was good for the TPS (ICC = 0.84; 95% CI 0.74–0.90, *p* < 0.001), moderate for the CPS (ICC = 0.73; 95% CI 0.57–0.83, *p* < 0.001), and poor for the ICS (ICC = 0.31; 95% CI 0.05–0.53, *p* = 0.01) (Figure 4B). The TPS and CPS did not differ significantly between human and machine (Wilcoxon test, *p* = 0.723 and *p* = 0.403), but differed significant for the ICS (Wilcoxon test, *p* < 0.001).

To verify the treatment-decisive concordance of automated PD-L1 scores with that of human investigators, we compared PD-L1 scores using typical cutoffs described in the literature. As cutoffs, we used ≥1% and ≥50% for the TPS, ≥1 and ≥20 for the CPS, and ≥1% and ≥5% for the ICS.

Human-machine concordance was best for TPS ≥ 50% with 94.4%, followed for CPS ≥20 with 90.7% and CPS ≥ 1 with 83.3%. Human-machine concordance was 77.8% for ICS ≥5% and 70.4% for ICS ≥ 1%. The worst concordance was observed with a TPS of ≥1% with 64.8% (Figure 5A–C). The weighted kappa between human and human was κ = 0.61 for the TPS, κ = 0.43 for the CPS, and κ = 0.27 for the ICS, and between human and machine it was κ = 0.34 for the TPS, κ = 0.58 for the CPS, and κ = 0.39 for the ICS. Overall, human-human concordance was better for the TPS compared between human and machine, but for the CPS and ICS weighted kappa was better for human-machine.

## 4. Discussion

In this study, we established and evaluated a fully automated PD-L1 scoring approach as PD-L1 scores are crucial for the selection of patients for PD-1/PD-L1 inhibitor therapy. We found different results in terms of comparing manual and automated PD-L1 scoring depending on the score assessed.

The ICC for the TPS between human and machine was good with a value of 0.84, but was surpassed with an excellent ICC of 0.91 between human and human. Nevertheless, the ICC for our automated TPS was consistent with the results of other studies in which Lin’s concordance correlation coefficient (CCC) between human and machine ranged from 0.84 to 0.95 for the TPS [16,33,36] and from 0.77 to 0.90 for the TPS grouped into six ordinal categories (<1%, 1–4%, 5 to 9%, 10% to 24%, 25% to 49%, ≥50%) [29]. It should be noted, that the ICC and CCC lead to almost identical coefficient and are thus comparable, however, the CCC can only be used for two raters [49]. Our concordance of 94% for a TPS cutoff of ≥50% was comparable to 92% from another study. However, for a TPS cutoff of ≥1%, our concordance of 64.8% was worse compared with 90% from that study [36], most likely due to artifacts discussed below. The concordance for the TPS cutoffs was notably better between human and human with κ of 0.61 than between human and machine with κ of 0.34. Overall, it should be noted that there were too few cases in the therapy-deciding range (TPS ≥ 50%) to allow a conclusion for clinical applications.

On the other hand, the ICC between human and machine for the CPS was moderate and very similar of 0.73 compared to human-human of 0.71. The concordance was 90.7% for a CPS cutoff of ≥20. For a low CPS cutoff of ≥1, this was 83.3% and almost identical to 83.6% reported by another study [32]. An ICC or CCC was not reported for the CPS by Kim et al. (2020), but it was shown that manual and automated CPS did not differ significantly in response to pembrolizumab (*p* = 0.186), suggesting that automated CPS could possibly replace a manually determined score in treatment decision-making [32]. This was also reflected in the concordance for the CPS cutoffs, which was notably better for human-machine (κ = 0.58) than that for human-human (κ = 0.43). Our kappa values were in the range to a multicenter inter-rater study for HNSCC, which reported a Fleiss’ kappa value between 0.303 and 0.557 for the CPS [51]. Overall, this supports the potential clinical value of machine-based decision finding for the treatment of HNSCC based on a CPS cutoff of ≥1.

Although ICS is not yet used in HNSCC, it is routinely determined in other tumor entities [14]. As we know this is the first time that the ICS has been determined in an automated manner. However, the agreement for both human-human and machine-human ICS was poor, with an ICC of 0.26 and 0.31, respectively. Concordance was slightly better at 77.8% for an ICS cutoff of ≥5% compared with 70.4% for a cutoff of ≥1%. Although we determined ICS (percentage of the area of PD-L1-positive infiltrating immune cells in respect to the total area of the tumor tissue) instead of IC% (proportion of membranous positive PD-L1 immune cells based on the total immune cell population) [29], which is therefore not equivalent to ICS by definition [10], we similarly observed that the automated determination of ICS such as IC% was worse than the TPS in agreement to human values [29]. In contrast to the TPS and CPS (*p* = 0.723, *p* = 0.403), humans tended to have significantly higher ICS values compared to the machine (*p* < 0.001; see Figure 3C or Appendix A). However, the concordance for the ICS cutoffs was notably better for human-machine with a κ of 0.39 than for human-human with a κ of 0.27. Although the ICS is not currently used for HNSCC, it has an impact on the CPS as it also is based on the amount of PD-L1 positive immune cells. However, clinical application could be considered for other entities, as different ICS cutoffs are applied there [14,15].

It should be mentioned that one of the investigators (B.P.) is not a pathologist but is experienced in the field of neuropathology and received extensive training in PD-L1 scoring. We additionally calculated the PD-L1 score without B.P. and noted that the ICC did not improve significantly (TPS 0.92 vs. 0.91, CPS 0.710 vs. 0.708, and ICS 0.09 vs. 0.289). Its values were within the range of the other investigators (see Appendix A).

The observed discrepancies between human and machine PD-L1 scores could be caused by a number of factors. Visual and cognitive traps can occur during manual evaluation and bias scores [52]. For example, visual traps lead to a false perception of cell and nucleus size (illusion of size), the strength of membrane and cytoplasmic staining (color assimilation, lateral inhibition) and cognitive traps lead to avoidance of extreme values and rounding in scoring (avoidance of extreme ranges, number preference) and are influenced by previous cases (context bias). Furthermore, the inability to perceive immune cells if they rarely occur in the tissue under investigation (search satisfaction) [52]. Although scoring is performed by a trained and experienced pathologist as the gold standard, manual scoring inevitably remains only qualitative or semiquantitative and is ultimately subjective [52], especially at low scores [53]. Based on our results, visual and cognitive traps might play a role especially in CPS, where immune cells are counted together with tumor cells, thereby making it more complicated in comparison to the TPS. But even more so in ICS, where instead of cell counts, the areas of scattered immune cells need to be estimated compared to the total tumor area [10]. This also aligns with the fact that ICS determination by humans is highly inconsistent among a large number of investigators [54]. Agreement in ICS scores across observers was described between 38% (antibody clone SP142) and 50% (antibody clone SP263) for two categories (ICS < 1%, ≥1%) [54]. In this regard, human subjectivity might affect the possibility of agreement between human and machine-based PD-L1 scoring [52]. Interestingly, our human-machine agreement results of the different PD-L1 scores strictly follows those of human-human and do not significantly outperform them (Figure 4). This raises the legitimate question of whether machine scores never perfectly match human scores, where subjective component could be a relevant factor. One way to overcome this subjective component and provide a ground truth could be to determinate PD-L1 mRNA levels by RNA sequencing [55].

Another explanation for the observed discrepancy between human and machine PD-L1 scoring could be artifacts such as structural damage, overlapping tissue, carbon pigments, and variations in staining. These could lead the machine to false results, whereas this is usually compensated by humans.

Indeed, our machine TPS values fluctuate slightly above those of humans at low scores (≥1%), while being underestimated at higher scores (Figure 3A or Appendix A). Fluctuation at low scores may be due to misidentified structures as positive cells or too low DAB threshold which is used as a parameter to distinguish between positive and negative cells. However, finding an ideal DAB threshold for our studied cases was not in the scope of this study. Therefore, we applied a DAB threshold already described in the literature. However, optimization of this DAB threshold could therefore lead to improved human-machine agreement. Nevertheless, increased variation between individual human investigators was also seen for a TPS cutoff of ≥1% (Figure 5A).

At the same time, the machine (mean TPS of 7.1%) achieved slightly lower TPS scores than the human (mean TPS of 8.9%), which could be explained by a slight underestimation at high automated TPS scores. This could be explained by the fact that intense DAB staining at high scores could lead to greater desaturation of hematoxylin during digital staining separation in these sections [56]. Thereby, it could have a negative impact on cell classification, which would then more frequently detect stromal cells instead of tumor cells, resulting in false-low TPS scores which would no longer exceed the cutoff of ≥50% (compare Figure 3A or Appendix A).

Other reasons for the discrepancy may be the applied approach. In most studies, PD-L1 scores were determined using pre-selected TMAs or image patches [26,28,29,31] or manually annotated WSIs [27,33,36]. However, manual annotation of tumor tissues in WSIs is time consuming and TMAs and image patches does not meet the gold standard of manual PD-L1 scoring in entire slides.

CNN represent the state of the art in medical image recognition and could be used in automated tumor annotation [25]. Therefore, we used the CNN “shufflenet” architecture for automated tumor annotation [37] since it is not computationally intensive while providing high accuracy in tumor detection [57]. It was able to annotate HNSCC in WSIs and achieved an overall tissue classification accuracy of 96.8% (Figure 2B) and was comparable to another study that performed automated tumor annotation in lung adenocarcinomas and achieved an accuracy ranging from 88.52% to 99.17% [34]. Neither our model nor that of Wu et al. [34] was able to exclude artifacts. Semi-automated approaches (TMAs, image patches, manually annotated WSIs) instead having the advantage that artifacts can be manually excluded or significantly reduced.

In addition to tumor annotation, cell detection may be another source of error. Surprisingly instead of state-of-the-art deep learning methods [42], in many applied methods, cell detection has been performed using the watershed algorithm [26,28,30,34,36], which was already described in the end of 1970s [58]. To achieve better cell detection results, an already trained CNN model “StarDist” was used in our approach (Figure 1B). It is named by the use of star-convex polygons to detect cell nuclei and competes with the state of the art in cell detection and has a high performance and more plausible results [42].

Although the other approaches mainly used random decision forests (RDFs) for cell classification (RDF) [27,30,36], a MLP was used instead for cell classification since, unlike RDFs, it would be combinable as CNN/MLP for future improved approaches [59]. MLPs are artificial neural networks whose functionality has no exact biological counterpart and have been used for classification tasks of cells [44]. Our MLP classified cells into tumor, immune and stromal cells and reached an overall accuracy of 97.2% (Figure 2C). A comparison with the results of the RDFs of the above-mentioned studies was not possible since no corresponding parameters (accuracy, variable importance, etc.) were given.

A completely different approach attempts to directly map PD-L1 scores using Deep Learning [16,35], which might allow it to be less vulnerable to artifacts. Using an auxiliary classifier generative adversarial network for tumor detection, the TPS could be calculated based on the ratio of the pixel numbers of positive and negative tumor areas. Generative adversarial networks basically consist of two neural networks (discriminator and generator), which compete with each other and allow the model to learn [16,35]. Although this deep learning approach was able to determine the TPS in entire WSIs, the interpretability of the underlying decision criteria was not provided. Deep learning methods are difficult to be interpretated [25]. To address the problem of the non-explainability of our model, we visualized the decision criteria for both tumor annotation (Figure 2A,B) and cell classification (Figure 2C,D) and thus made them explainable to provide confidence in our approach.

Likewise, proprietary software was often used [29,32,33], making third-party verification difficult or impossible compared to open-source software such as QuPath. By combining MATLAB and QuPath, our approach can be easily replicated by third parties. The use of Python instead of MATLAB would allow for a fully open-source solution.

However, the following limitations should be considered. First, automated PD-L1 scoring depends on good performance in tumor detection, cell detection and classification. Poor performance in this area may add up and lead to altered PD-L1 scores. Second, this study used a DAB threshold for optical density already described in the literature and could have an inverse impact on our results. In addition, a pre-selection of WSIs showing extensive artifacts should be performed by a human (e.g., technical assistant) that could negatively affect automated PD-L1 scoring. Moreover, automated tumor detection and tumor cell recognition was model-specific for HNSCC since the training data were based on HNSCC (CPTAC). However, the models could likely be applied to other squamous cell carcinomas without adaptation. Nevertheless, application to other entities would require retraining for tumor tissue detection (1st neural network) and tumor cell classification (3rd neural network). Overall, the tool presented here differs from other tools in that it is able to automatically detect invasive carcinoma areas and determine all three PD-L1 scores in WSIs simultaneously. As an open-source software, it can be tested, modified and, most importantly, further developed by anyone (See Appendix A).

## 5. Conclusions

Consecutive combination of neural networks enables automated PD-L1 scoring of all three PD-L1 scores based on WSIs of PD-L1 IHC in HNSCC cases and hematoxylin-based detection of HNSCC tumor tissue not yet described in literature. Our study provides deeper insights into automated PD-L1 scoring by neural networks and its limitations and can serve as a basis to obtain reliable and reproducible PD-L1 scores for patient selection for PD-1/PD-L1 immune checkpoint inhibitor therapy in the future. Further efforts should be made to reduce artifacts in automated PD-L1 scoring and provide methods to compare both human and machine PD-L1 scores with a ground truth.

## Figures and Tables

**Figure 1 cancers-13-04409-f001:**
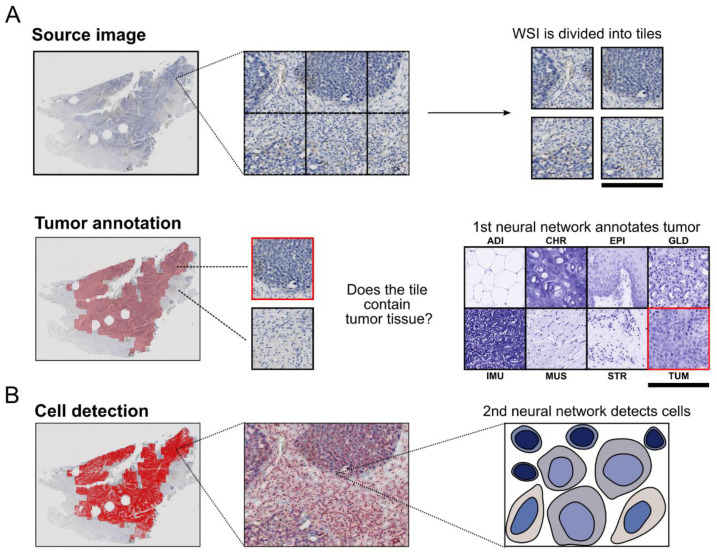
Method for the automated determination of PD-L1 scores. (**A**–**D**) A Whole slide image (WSI) from routine diagnostics. Note that the holes in the tissue correspond to sampling sites of previously made TMAs. (**A**) Step 1: The whole slide image is first divided into separate tiled images. Then, using the first neural network (CNN “shufflenet”), each tile image is classified into one of eight tissue classes (ADI, adipose tissue; CHR, cartilage tissue; EPI, epithelium; GLD, glandular tissue; IMU, immune cells; MUS, muscle tissue; STR, stromal tissue; TUM, tumor tissue) in a sliding window manner. The regions of WSI whose tile images were classified as tumor are annotated. Scale bar equals 256 µm. (**B**) Step 2: In the annotated tumor region, the cells are detected and quantified (cell/nucleus size and shape, strength of hematoxylin staining and DAB staining, etc.) by the second neural network (CNN “StarDist”). (**C**) Step 3: All recognized cells are classified by the third neural network (MLP) into tumor, immune or stromal cells based on their characteristics. (**D**) Step 4: Positive tumor (based on the membrane) and immune cells (based on the whole cell) are detected using a DAB optical density threshold. Thereby, the TPS, CPS and ICS can be finally calculated.

**Figure 2 cancers-13-04409-f002:**
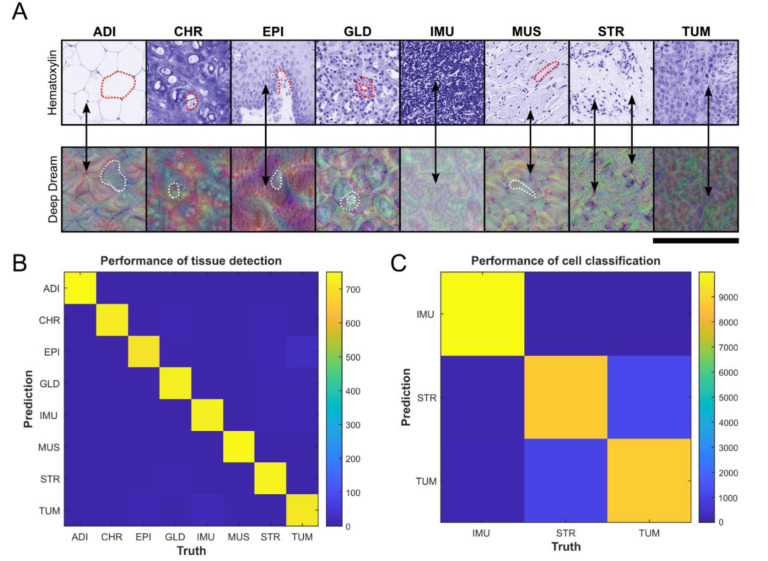
Neural Network Performance and Deep Dream. (**A**) Hematoxylin-stained tile images of the 8 tissue classes are shown in the top row. In the bottom row, abstractions of histological tissue classes are visualized using “Deep Dream” method, which the trained CNN “shufflenet” can recognize: ADI (adipose tissue): Vesicular structures and star-shaped retractions corresponding to vacuoles and nuclei; CHR (cartilage tissue): Typical configuration of chondrocytes; EPI (epithelium): Stratification of cells based on a basal lamina; GLD (glandular tissue): Cloudy structure corresponding to glands with a central lumen; IMU (immune cells): Grape-shaped collections of cells. MUS (muscle tissue): Muscle fibers both transversely and longitudinally sectioned with associated nuclei. STR (stromal tissue): Single cell nuclei embedded in a network of collagen fibers; TUM (tumor tissue): Cells of different sizes without recognizable basal lamina. Important structures are shown by dotted lines (hematoxylin: In red; deep dream: in white). The black arrows point to individual cells in hematoxylin and deep dream images. Scale bar equals 256 µm. (**B**) Accuracy of the 1st neural network (CNN “shufflenet”) based on 6000 tile test images (750 per class). (**C**) Accuracy of the 3rd neural network (MLP) based on 30,000 detected cells (normalized to 10,000 per class). (**B**,**C**) The *x*-axis shows the true class and the *y*-axis shows the prediction. (**D**,**E**) Importance variables of the third neural network (MLP) (Olden et al. 2004). (*y*-axis): the variables correspond to morphological or color properties of the recognized cells. (*x*-axis): raw and smoothed data with 25 µm. A strong yellow or green color corresponds to a high importance variable. Nearby detection counts exists only for smoothed data.

**Figure 3 cancers-13-04409-f003:**
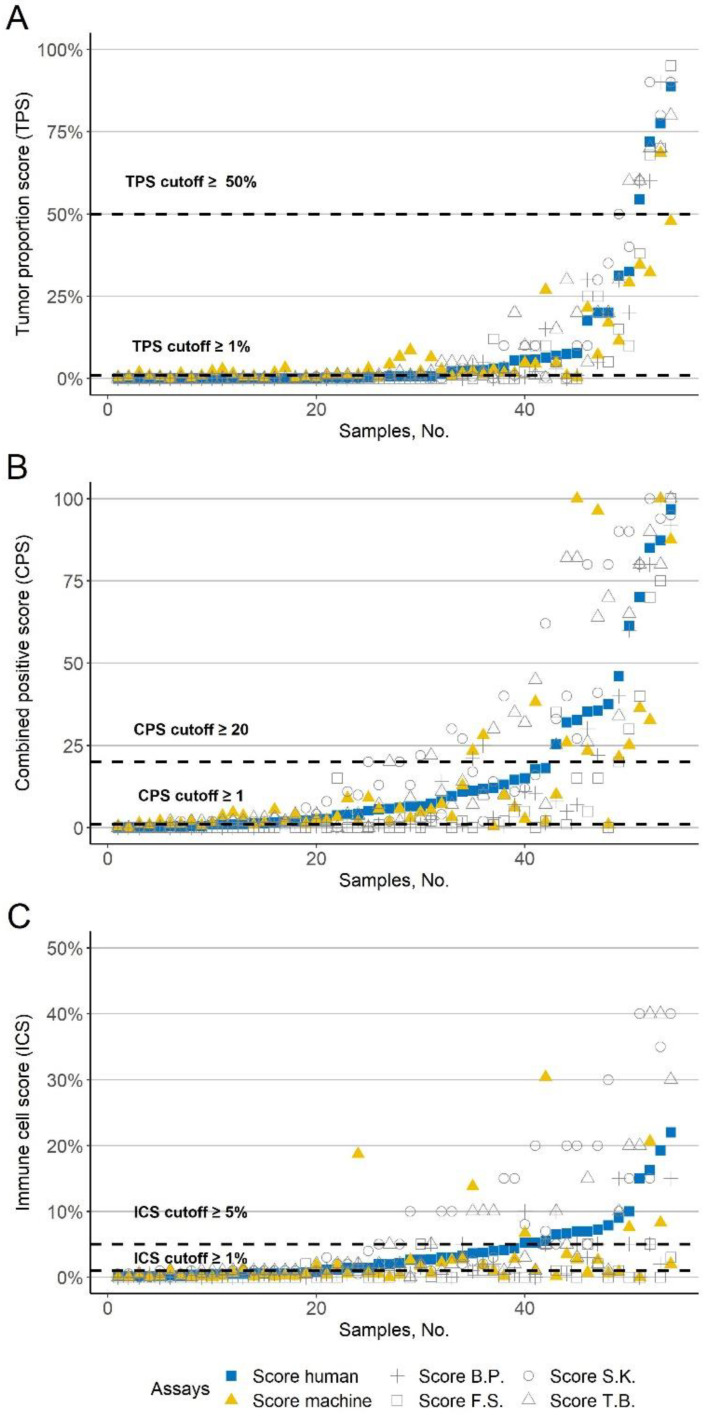
Results of PD-L1 scoring. (**A**–**C**) Three PD-L1 scores for all cases. 
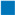
 PD-L1 average score of the four human investigators. 
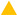
 Automated PD-L1 score based. The gray symbols correspond to the four investigators (+ B.P., □ F.S., ○ S.K., and △ T.B.); The *x*-axis shows the samples. The *y*-axis shows the corresponding PD-L1 scores. The dashed line indicates common cutoffs. The values are sorted in ascending order according to the PD-L1 average score of the four investigators. (**A**) Tumor proportion score (TPS). (**B**) Combined positive score (CPS). (**C**) Immune cell score (ICS).

**Figure 4 cancers-13-04409-f004:**
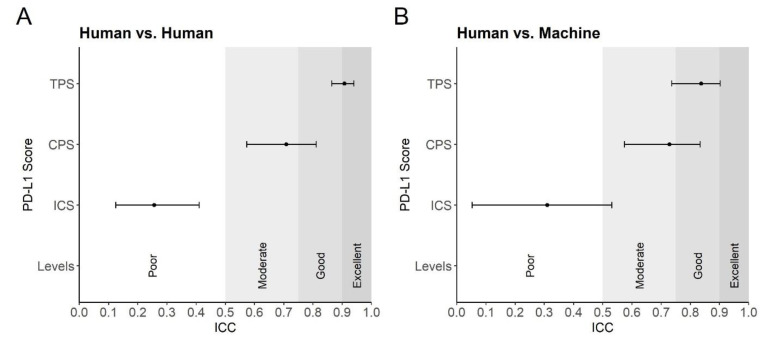
Intraclass correlation coefficient for PD-L1 scores. (**A**,**B**) The interclass correlation coefficients (ICCs) for TPS, CPS and ICS. The ICC is shown on the *x*-axis. An ICC of 0 to 0.5 is rated as poor, 0.5 to 0.75 is rated as moderate, 0.75 to 0.9 is rated as good, and 0.9 to 1.0 is rated as excellent. The *y*-axis shows the three PD-L1 scores (TPS, CPS, ICS) and “Levels” represents the interpretation levels of the ICC in words. The dots correspond to the estimated ICC and the horizontal bar represents the confidence interval; (**A**) PD-L1 scores compared between the four human investigators; (**B**) Automated PD-L1 scores were compared with manual PD-L1 scores.

**Figure 5 cancers-13-04409-f005:**
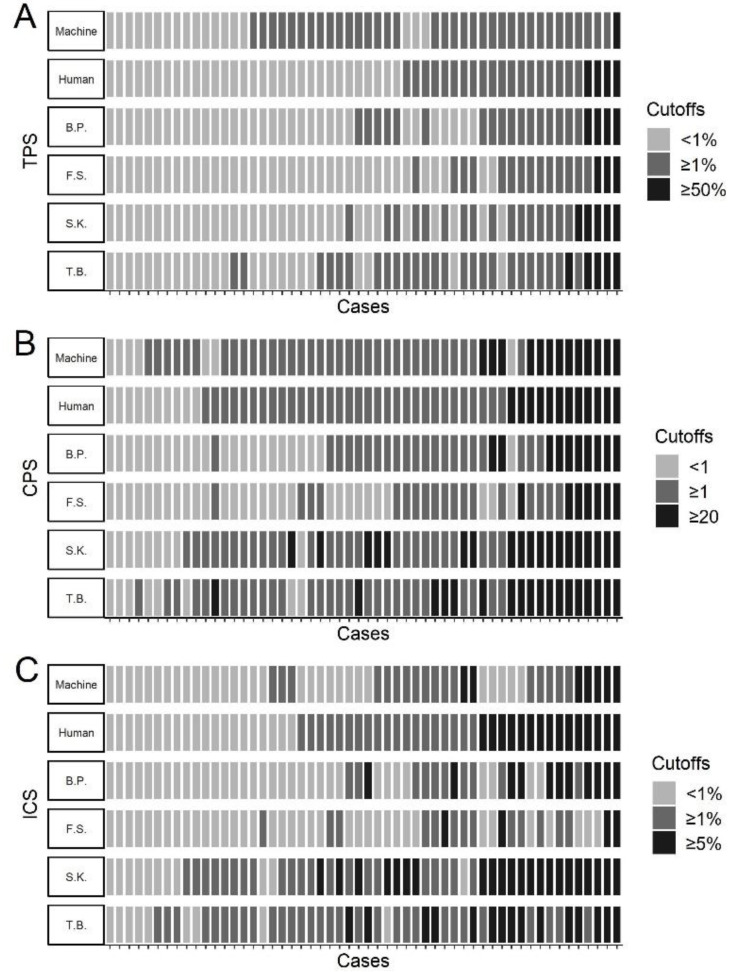
Agreement of PD-L1 scores by cutoffs. (**A**–**C**) Individual cases are shown as boxes on the y axis (Machine = automated PD-L1 scoring, Human = mean of the four human investigators, B.P., F.S., S.K. and T.B. = the four investigators in this study); (**A**) Interrater agreement of TPS. Cutoffs of <1%, ≥1%, and ≥50% were chosen; (**B**) Interrater agreement of CPS. Cutoffs of <1, ≥1, and ≥20 were chosen; (**C**) Interrater agreement of ICS. Cutoffs of <1%, ≥1% and ≥5% were chosen.

## Data Availability

The data presented in this study are available on reasonable request from the corresponding author. The source code used is available in Appendix A.

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
