# Peer review of "Automated PD-L1 Scoring Using Artificial Intelligence in Head and Neck Squamous Cell Carcinoma"

_cancers, 2021, doi:10.3390/cancers13174409_

Round 1

Reviewer 1 Report

The study by Puladi et al. deals with an automatic scoring of PDL1 immunostaining in head and neck squamous cell carcinoma (HNSCC). For this purpose, the authors use three consecutive neural networks. In the first step, the neural network ("shufflenet") should annotate the WSI into the eight classes: tumor, stroma, immune cells, adipose tissue, cartilage, epithelium, glands and muscles. Then the cells are to be recognized with their cell boundaries (CNN, "StarDist") and in the last step the cells are then classified as either tumor, stromal or immune cells (ANN-MLP).

This procedure corresponds to the current standard of technology and is well suited and worked out for the current study.

However, before the actual cases can be evaluated, the individual networks must be trained and validated according to their task. In the current study, this is done with a public data set (CPTAC-HNSCC).

Using suitable statistical methods, the results are compared with conventional evaluation by trained pathologists.

 At the same time, the results are critically discussed against the background of partly discrepant evaluations between machine and human. This is done in a scientific and sufficient manner. The manuscript is very well structured and easy to read. The working hypothesis was rigorously worked through and the methods used in accordance with the current standard.

However, before the manuscript can be published, the authors should critically adress the following:

- The specified public collective is very difficult to find at the specified address. If it is about the CPTAC-HNSCC collective, the following address would be recommended for direct access: https://wiki.cancerimagingarchive.net/display/Public/CPTAC-HNSCC

- For better reproducibility, the code used should either be uploaded to a corresponding repository (e.g. Github) and linked in the text or included in the supplementary files. This also makes it easier for the reader to understand the results better.

- The processing of the WSI (e.g. how the tiles were created etc.) should be described in more detail in the material and methods section.

- The text should be checked again carefully for spelling mistakes.

Author Response

We uploaded the comments also as a word file.

Dear Editor and Dear Reviewers,

Thank you very much for taking the time to review our manuscript and for providing valuable feedback and comments.

Please find below a response to each reviewers’ comment. All changes made to the text (as uploaded separately) are highlighted.

Best regards,

Till Braunschweig

Senior Pathologist, corresponding author

Institute of Pathology,

University Hospital RWTH Aachen, Germany

[email protected]; Tel.: +49-241-80-89282

Reviewer #1

The study by Puladi et al. deals with an automatic scoring of PDL1 immunostaining in head and neck squamous cell carcinoma (HNSCC). For this purpose, the authors use three consecutive neural networks. In the first step, the neural network ("shufflenet") should annotate the WSI into the eight classes: tumor, stroma, immune cells, adipose tissue, cartilage, epithelium, glands and muscles. Then the cells are to be recognized with their cell boundaries (CNN, "StarDist") and in the last step the cells are then classified as either tumor, stromal or immune cells (ANN-MLP).

This procedure corresponds to the current standard of technology and is well suited and worked out for the current study.

However, before the actual cases can be evaluated, the individual networks must be trained and validated according to their task. In the current study, this is done with a public data set (CPTAC-HNSCC).

Using suitable statistical methods, the results are compared with conventional evaluation by trained pathologists.

 At the same time, the results are critically discussed against the background of partly discrepant evaluations between machine and human. This is done in a scientific and sufficient manner. The manuscript is very well structured and easy to read. The working hypothesis was rigorously worked through and the methods used in accordance with the current standard.

 However, before the manuscript can be published, the authors should critically address the following:

 - The specified public collective is very difficult to find at the specified address. If it is about the CPTAC-HNSCC collective, the following address would be recommended for direct access: https://wiki.cancerimagingarchive.net/display/Public/CPTAC-HNSCC

We are grateful for the information and have included the mentioned link in the manuscript.

- For better reproducibility, the code used should either be uploaded to a corresponding repository (e.g. Github) and linked in the text or included in the supplementary files. This also makes it easier for the reader to understand the results better.

We thank the reviewer for the recommendation and will upload a supplementary file with the source code on MDPI.

- The processing of the WSI (e.g. how the tiles were created etc.) should be described in more detail in the material and methods section.

We thank the reviewer for that advice and have added more detailed information in the material and methods section for the processing of the WSIs.

- The text should be checked again carefully for spelling mistakes.

Following this advice, we had reviewed the manuscript by a native speaker and corrected the text accordingly.

Reviewer 2 Report

Overall an interesting studying looking at an automated method for cell type detection with PD-L1 scoring.  The method demonstrates compelling method for whole slide evaluation for automation of scoring.  Overall it appears that automated methods perform similarly to human evaluation and therefore could be considered for clinical use.  Main concerns are that at low cut off scores, accuracy is lower, and these differences are clinically relevant.  For instance, FDA & EMA approval for pembrolizumab for recurrent/metastatic tumors is for CPS score >=1.  Misclassification in this lower scoring range can have major clinical implications and this should be discussed as whether it is a limitation to transitioning to clinical use.

Why the focus on head and neck tumors? Do these tumors have specific features that make them easier or harder to grade than other tumor types?  Are there features of the model that are tumor type specific?  Discussion of the clinical implications for head and neck tumors should be included.  Also discussion on whether this could be applied to other tumor types should be included

Why is one rater not a pathologist?  Can you show that this rater did not have significantly different evaluation or deviation is not greater than the other pathologists?

For cut off scores (more clinically relevant end point) can you compare human- human concordance with human-machine concordance? 

How was the human cut-off score determined? Average of all human scores?

Can you highlight how this tool is different than other tools that have been previously published in the discussion more? Compared to the tools that this is being compared to in the discussion. Because it is from whole slide and does all three scores?

A bit more reference on relevance of ICS score should be included in discussion.  Would include some clinical references as to whether this score is clinically relevant.  Most clinical studies use TPS or CPS https://pubmed.ncbi.nlm.nih.gov/30678715/

Figure 3 – consider log scale (cannot see performance for lower scores very well)

Figure 3 - Graphs need legends on the side (not just in the caption)

Some of the wording could be improved “Out of nowadays” in the intro for instance

Author Response

We uploaded simultaneously a word file with the below text and figures.

Dear Editor, Dear Reviewers,

Thank you very much for taking the time to review our manuscript and for providing valuable feedback and comments.

Please find below a response to each reviewers’ comment. All changes made to the text (as uploaded separately) are highlighted.

Best regards,

Till Braunschweig

Senior Pathologist, corresponding author

Institute of Pathology,

University Hospital RWTH Aachen, Germany

[email protected]; Tel.: +49-241-80-89282

Reviewer #2

Overall an interesting studying looking at an automated method for cell type detection with PD-L1 scoring.  The method demonstrates compelling method for whole slide evaluation for automation of scoring.  Overall it appears that automated methods perform similarly to human evaluation and therefore could be considered for clinical use.  Main concerns are that at low cut off scores, accuracy is lower, and these differences are clinically relevant.  For instance, FDA & EMA approval for pembrolizumab for recurrent/metastatic tumors is for CPS score >=1.  Misclassification in this lower scoring range can have major clinical implications and this should be discussed as whether it is a limitation to transitioning to clinical use.

Why the focus on head and neck tumors? Do these tumors have specific features that make them easier or harder to grade than other tumor types?  Are there features of the model that are tumor type specific?  Discussion of the clinical implications for head and neck tumors should be included.  Also discussion on whether this could be applied to other tumor types should be included

We thank the reviewer for this valuable feedback. The focus of our research group (pathologists and surgeons) is the diagnosis, therapy and molecular background of head and neck squamous cell carcinoma including aspects of immuno-oncology and immunotherapy. To our knowledge, automated PD-L1 assessment for squamous cell carcinoma of the head and neck has not yet been described. Therefore, we were interested in evaluating automated PD-L1 scoring using HNSCC.

As for tumor type characteristics, HNSCC have a very heterogeneous tumor microenvironment, which complicates PD-L1 evaluation and ultimately treatment decisions. Automated tumor detection and recognition of tumor cells is model specific for HNSCC, because training data were based on HNSCC. However, the models could probably be performed on other squamous cell carcinomas without adaptation. But, for the application to other entities, retraining for tumor tissue detection (1st neural network) and tumor cell classification (3rd neural network) would be required.

The clinical implication was critically discussed for each PD-L1 score and added to the manuscript.

Why is one rater not a pathologist?  Can you show that this rater did not have significantly different evaluation or deviation is not greater than the other pathologists?

We understand reviewers’ concerns.

The investigator, B.P., conducted his doctoral thesis in the field of neuropathology. See (https://www.sciencedirect.com/science/article/pii/S0969996120304137 and https://www.sciencedirect.com/science/article/abs/pii/S0006899318305602). Thus, he has sufficient prior knowledge of histopathology to include him as an additional investigator for PD-L1 scoring after intensive training on test cases.

However, to address the mentioned concerns, we added a Supplementary Figure 2 comparing the results of all four investigators. Scores by B.P. (mean and standard deviation) are in range of the other investigators.

Furthermore, we calculated the ICC without B.P., which did not improve significantly. For the TPS we had 0.92 instead of 0.91, for the CPS we had 0.710 instead of 0.708, and for the ICC we even had a decline to 0.09 instead of 0.289. We additionally addressed these concerns in the discussion with a sentence.

For cut off scores (more clinically relevant end point) can you compare human- human concordance with human-machine concordance? 

Thank you for this good advice. We added a weighted kappa concordance in the manuscript (since it is ordinal data).

How was the human cut-off score determined? Average of all human scores?

Correct, the cutoff score was determined as the average of all four investigators‘scores.

Can you highlight how this tool is different than other tools that have been previously published in the discussion more? Compared to the tools that this is being compared to in the discussion. Because it is from whole slide and does all three scores?

We thank the reviewer for this comment and consideration. This tool differs from other tools in that it detects invasive carcinoma automatically and can determine all three PD-L1 scores in WSIs fully automated and simultaneously. By making it available in a supplementary file on MDPI, it can be tested, modified and, above all, further developed by anyone.

A bit more reference on relevance of ICS score should be included in discussion.  Would include some clinical references as to whether this score is clinically relevant.  Most clinical studies use TPS or CPS https://pubmed.ncbi.nlm.nih.gov/30678715/

We thank the reviewer for this comment and suggestion. The ICS provides useful insight about the amount of PD-L1 positive immune cells. Since the CPS is based on a combination between PD-L1 positive immune and tumor cells, we therefore wanted to also investigate the ICS. Furthermore, atezolizumab has been approved in triple-negative breast cancer for an ICS of >1% and in urothelial carcinoma for an ICS of ≥5%. Therefore, it might be of value in future therapeutic approaches. We mentioned this in the introduction and discussion.

Figure 3 – consider log scale (cannot see performance for lower scores very well)

We thank you very much for the advice. For better visualization of low scores, we added additional representation with pseudo-log scale as Supplementary Figure 1 (see below).

Figure 3 - Graphs need legends on the side (not just in the caption)

We thank the reviewer for the advice. We have added a legend to Figure 3 and the supplementary Figure 1 and allowed ourselves to position it at the bottom. We hope that you agree with this solution.

Some of the wording could be improved “Out of nowadays” in the intro for instance

Following this advice, we had reviewed the manuscript by a native speaker and corrected the text accordingly.
